# Hydrogen Production from Coffee Mucilage in Dark Fermentation with Organic Wastes

**Edilson León Moreno Cárdenas** [1,†], **Arley David Zapata-Zapata** [2,†] and **Daehwan Kim** [3,*]

1 Laboratorio de Mecanización Agrícola, Departamento de Ingeniería Agrícolay de Alimentos, Universidad Nacional de Colombia-Sede Medellín, Carrera 64c # 63-120, Código Postal 050034, Colombia; elmorenoc@unal.edu.co

2 Universidad Nacional de Colombia-Sede Medellín-Escuela de Química-Laboratorio de Procesos Biológicos-Carrera 65 # 59A-110, Medellín, Código Postal 050034, Colombia; adzapata@unal.edu.co

3 Department of Biology, Hood College, 401 Rosemont Avenue, Frederick, MD 21701, USA

\* Correspondence: kimd@hood.edu; Tel.: +1-765-637-8603

† Authors contributed equally to the study.

**Abstract:** One of primary issues in the coffee manufacturing industry is the production of large amounts of undesirable residues, which include the pericarp (outer skin), pulp (outer mesocarp), parchment (endocarp), silver-skin (epidermis) and mucilage (inner mesocarp) that cause environmental problems due to toxic molecules contained therein. This study evaluated the optimal hydrogen production from coffee mucilage combined with organic wastes (wholesale market garbage) in a dark fermentation process. The supplementation of organic wastes offered appropriate carbon and nitrogen sources with further nutrients; it was positively effective in achieving cumulative hydrogen production. Three different ratios of coffee mucilage and organic wastes (8:2, 5:5, and 2:8) were tested in 30 L bioreactors using two-level factorial design experiments. The highest cumulative hydrogen volume of 25.9 L was gained for an 8:2 ratio (coffee mucilage: organic wastes) after 72 h, which corresponded to 1.295 L hydrogen/L substrates (0.248 mol hydrogen/mol hexose). Biochemical identification of microorganisms found that seven microorganisms were involved in the hydrogen metabolism. Further studies of anaerobic fermentative digestion with each isolated pure bacterium under similar experimental conditions reached a lower final hydrogen yield (up to 9.3 L) than the result from the non-isolated sample (25.9 L). Interestingly, however, co-cultivation of two identified microorganisms (*Kocuria kristinae* and *Brevibacillus laterosporus*), who were relatively highly associated with hydrogen production, gave a higher yield (14.7 L) than single bacterium inoculum but lower than that of the non-isolated tests. This work confirms that the re-utilization of coffee mucilage combined with organic wastes is practical for hydrogen fermentation in anaerobic conditions, and it would be influenced by the bacterial consortium involved.

**Keywords:** hydrogen; coffee mucilage; organic wastes; dark fermentation; anaerobic digestion

## 1. Introduction

Conventional fossil fuels, the main energy sources for industrial/technological development, have been meeting about 80% of the fundamental energy demand and supply in the world [1–3]. However, this fossil fuel-dependent energy system causes problems of limited resources, greenhouse gas emissions, and environmental pollution issues [4–6]. Among diverse alternative clean energy resources, hydrogen has been considered as a prospective future energy source for replacing the gradual depletion of fossil fuels and addressing the lack of sustainability. Hydrogen energy is not only unrestricted by greenhouse gas emissions but also produces more than 2.5 times higher energy

than the energy from hydrocarbons [7,8]; this ability of hydrogen energy is proposed for an essential renewable and energy source for transportation. The current physico-chemical practice in most facilities achieves approximately 90% yields of hydrogen through steam refining of methane (40%), and gasification/partial oxidation of crude oil and coal (48%). However, these are still highly dependent upon fossil fuels because of cost effectiveness and the lack of a suitable alternative technique [1]. With these concerns about rapid depletion of petroleum stores and environmental problems, biological processes have been identified as a promising technology for hydrogen production. Its basic concept is to catalyze water decomposition or digest organic compounds in an environmentally friendly way using microorganisms, such as algae, *Cyanobacteria*, or photosynthetic bacteria. Biological methods can be classified into four groups: (1) direct bio-photolysis, (2) indirect bio-photolysis, (3) photo fermentation, and (4) dark fermentation. Although photodecomposition methods result in relatively higher hydrogen yields than the other approaches, the anaerobic fermentative process, in particular, dark fermentation, is widely thought to be an attractive approach. Since various organic wastes and wastewater can be used as carbohydrate-rich substrates in the anaerobic digestion process, and it is capable of transforming organic wastes into value-added molecules without light sources [9–11].

　　There is no doubt that coffee is one of the most largely consumed beverages along with water and tea worldwide; millions of people around the globe consume coffee each day, and the number of coffee-consuming people and nations are increasing. It is known that coffee is ranked number two as a traded commodity only after crude oils, with a worldwide production of coffee is estimated to be 152 million 60 kg bags [12]. Over the past decade, coffee production and its associated market have been rapidly growing with attractive research in functional foods, for example, the cognitive and physical behavior effects of caffeine. As coffee production and consumption increases, large amounts of undesirable byproducts (skin, parchment, pericarp, pulp, and mucilage) are also generated during the coffee separation process. In general, only coffee beans are used for brewed coffee, but the other components are separated and removed; they constitute more than 50% of an initial coffee fruit weight [13]. The residual coffee wastes after the wet separation process include 43.2% ($w/w$) skin and pulp, 6.1% ($w/w$) parchment, and 11.8% ($w/w$) mucilage and solubles [14]. Mussato et al. [15] reported the generation of residual wastes from the preparation of instant coffee were around 6 million tons per year worldwide, while more recent work has estimated the coffee residual byproducts would be approximately 15 million tons per year [16]. Coffee wastes can be utilized in animal feed [17–19], manure [20], antioxidant polyphenols [21,22], adsorption molecules [23–25], $\alpha$-amylase [26], and ethanol production [17,19,27]; however, most coffee wastes are unutilized and dumped into land or water for economical and/or technical reasons [13,16]. Considering the current facts and issues, further investigations and practical applications with coffee residue by-products are required.

　　Coffee mucilage is a colorless thin layer, mainly composed of water, sugars, protein, and pectin that covers the parchment and outer skin (pericarp), and protects the inner fibrous pulp and endosperm (coffee bean) components. Due to the high carbohydrate and nitrogen content in mucilage, it is one of the direct resources for animal feeds after agricultural processing and can be a suitable source of value-added molecules, such as ethanol, lactic acid, and hydrogen. Previous works identified that the coffee mucilage contained 85–91% ($w/w$) water, 6.2–7.4% ($w/w$) sugars, 4–5% ($w/w$) protein, and 1% ($w/w$) pectin substances [13,20,28,29], which were relatively higher than those obtained from other coffee by-products such as husks, skin, and pulp [19,30]. Furthermore, the sugars in the coffee mucilage include a high portion of reducing sugars (63%, $w/w$) that facilitates the utilization of sugars to other molecules and commodities [17,31]. Orrego et al. [13,16] reported that coffee mucilage from the wet separation process had >50 g/L of fermentable sugars (mainly glucose and galactose), acetic acid, protein, and some minerals (calcium, iron, magnesium, potassium, phosphorus, and sodium), which could be directly transformed into other molecules (e.g., ethanol), without requiring any pretreatment and carbon or nitrogen source supplements.

　　In order to develop the hydrogen process, several research studies have been conducted in the anaerobic digestion process in the presence of pure culture medium using a specific microorganism

such as *Clostridium*, *Bacillus*, and *Thermoanaerobacterium* [9,32,33]. However, hydrogen production under special conditions using defined/pretreated culture medium with a pure microbial inoculum restricts further understanding of the co-cultivation, substrate changes, biochemical and molecular interactions of the bacterial population into substrates and their patterns. Various seeding substrates, rich in carbohydrates, such as wastewater, sludge, compost, manure, and soil, are acceptable sources for fermentative hydrogen production, while to the best of our knowledge, there has been no work regarding optimization of hydrogen production from coffee mucilage in the dark fermentation method with organic wastes. This work reports that the use of coffee mucilage combined with supplemental organic wastes can be a potential approach for hydrogen production. The main objective of this study is to determine the effective practical conditions for fermentative digestion of organic compounds into hydrogen, which are evaluated at 30 L bioreactors for different ratios of coffee mucilage and organic wastes without inoculating any microorganism. The maximal cumulative hydrogen is achieved at a two-level factorial experimental design, and further tests are carried out and compared with different independent factors (ratio of coffee mucilage and organic wastes, chemical oxygen demand, pH, and temperature). Moreover, the impact of microbial consortiums (bacterial populations) on hydrogen production are tested with isolated bacteria under similar experimental conditions and those results were compared to non-isolated fermentation.

## 2. Materials and Methods

### 2.1. Raw Materials

Coffee mucilage was supplied by the San Rafael farm (Antioquia, Colombia), located at 1575 m above sea level with an average temperature of 21 °C. Organic wastes were collected from the Central Mayorista de Antioquia (Antioquia's Wholesale Market, Medellín, Colombia), any mainly contained fruit and vegetable wastes (lettuce, orange, guava, mango, and papaya), not suitable for human consumption (expired products). As soon as the raw materials were obtained, the coffee mucilage was autoclaved at 121 °C for 15 min and stored with intact organic wastes at 4 °C. The large solids in the mucilage sample were sieved over a 20-mesh screen (0.84 mm, Tyler USA standard testing sieve, VWR, Philadelphia, PA, USA), and the resulting slurry was centrifuged at 8000 rpm at 5 min in order to separate the remaining solids. Sugars and the acetic acid content of the mucilage liquid was determined by HPLC in our previous study, including glucose (37.1 g/L), galactose (14.7 g/L), lactose (0.8 g/L), and acetic acid (1.2 g/L), respectively [13]. All other chemicals and reagents in this study were purchased from Sigma Aldrich (St. Louis, MO, USA).

### 2.2. Experimental Design

Anaerobic fermentative digestion was prepared and evaluated by the Minitab 16 software program (Minitab 16, Minitab Inc., State College, PA, USA) with a two-level factorial experimental design. Initially, two prepared coffee mucilage and organic wastes were blended into three different ratios (*w/w*) of 8:2, 5:5, and 2:8, which were named level 1, level 2, and level 3, respectively. Two more levels with only coffee mucilage (level 0) or organic wastes (level 4) were added as control tests. Each anaerobic batch fermentation was carried out in a 30 L bioreactor with a working volume of 20 L under given experimental conditions: temperature range of 30–40 °C, chemical oxygen demand (COD) range of 20 g oxygen/L–60 g oxygen/L, and pH range of 5.0–8.0 until the hydrogen production was completed. The COD was determined by the 5220D Standard Chemical Oxygen Demand Method [34,35], and the initial pH was adjusted by adding 2 M of NaCl or NaOH. The quantity of total solids and volatile solids were determined following the previous work [36]. The bioreactor was operated with a helical ribbon impeller mixing at 100 rpm to avoid deposition of solids and to enhance the hydrogen turnover to the gas phase. The desirable temperature during the fermentation was kept by a heating jacket equipped with a main system, which was recorded with an automatic thermometer of 1 °C resolution and an accuracy of ±1 °C. The biogas samples from fermentative

digestion were collected in 1 L gas sampling Tedlar bags (model number: 22,950, Restek, Los Angeles, CA, USA) every 24 h.

### 2.3. Isolation, Gram Staining, Biochemical, Sequence Analysis of Microorganisms

In some experiments with high hydrogen production, liquid samples were taken after fermentation performances for further analyses for microbial identification, biochemical tests, sequence analysis, acids, and fermentation tests with pure bacterium. The liquid broth samples were bottled in sterilized jars and stored at 4 °C prior to use. Each sample was shaken for 2 min in a shaking incubator at 200 rpm, and cell concentration was adjusted to a $10^{-8}$ (Colony-forming unit) CFU/mL by a serial dilution with sterilized water. Each diluted aliquot was spread out on nutrient agar medium (0.5% peptone, 0.3% yeast extract, 1.5% agar, and 0.5% sodium chloride) and cultured at room temperature for a week. Each grown colony was picked, diluted in 0.5 mL of distilled water, and kept with 50% glycerol solution at −80 °C prior to further use.

In order to a microbial identification, the colorimetric identification card method was prepared via a compact Vitek2 device (Biomerieux, Lyon, France) equipped with a reactive card for biochemical tests according to previous work [37]. Briefly, each isolated microorganism was suspended with a sterilized saline solution (containing 0.5% NaCl, pH 7.0) until turbidity between 0.5 and 0.63 units on the McFarland scale (approximately cell density between $1 \times 10^8$–$1.89 \times 10^8$). The suspended cells (3 mL) on the identification card were installed in the Vitek2 device, and each sample was incubated at 35 °C for 12 h. After incubation, biochemical reactions were carried out with the values from the device's database, providing appropriate results based on the reactions.

In order to verify unknown microorganisms obtained from the best anaerobic fermentative digestion, the ribosomal DNA (16S rDNA) of each microorganism was prepared and isolated by the FastDNA spin kit for soil DNA (VWR catalog number: ICNA116560200, VWR Scientific, Bridgeport, NJ, USA). The 16S rDNA sequence was amplified through the polymerase chain reaction (PCR) with two designed primers: forward prime 5′-AGAGTTTGATCCTGGCTCAG-3′ and reverse prime: 5′-GGTTACCTTGTTACGACTT-3′. The polymerase chain reaction step was conducted in a Perkin-Elmer thermal cycler (GeneAmp PCR System 9700, Norwalk, CT, USA) 30 times. Each cycle included the denaturation step at 95 °C for 45 s, the annealing step at 56 °C for 2 min, and the extension step at 72 °C for 3 min [38]. The amplified PCR products were purified using the Wizard PCR preps DNA purification system (catalog number: A7231, Promega Corporation, Madison, WI, USA). The sequences of both directions of the DNA was confirmed via the ABI PROSM 3700 DNA analyzer (Applied Biosystem, Midland, ON, Canada) and the sequence alignment was carried out with BLAST at the NCBI.

For further dark fermentation with isolated pure bacterium, each isolated cell was grown overnight in a 500 mL Erlenmeyer flask (Belloco, Vineland, NJ, USA) in the presence of a YEPD medium (1% yeast extract, 1% peptone, and 2% glucose) at 30 °C with 200 rpm. The cells were harvested by centrifugation (5 min, 8000 rpm) then were suspended in YEP (no glucose) medium [39]. This liquid was utilized to inoculate the single cell fermentation with an initial cell concentration of 1 g dry cells/L. Each run was tested in a 30 L bioreactor (20 L working volume) under similar experimental conditions until the hydrogen production was completed and compared to the results from those from co-cultivations. All fermentation tests were conducted in duplicate.

### 2.4. Analysis

Hydrogen gas was analyzed using gas chromatography (3000 MicroGC system, Agilent, San Jose, CA, USA) equipped with a thermal conductivity detector (TCD) and capillary HP-PLOT U column (0.32 mm ID × 8 m length × 10 μm film). The temperatures of the injector, column, and detector were operated at 60 °C, 80 °C, and 300 °C respectively. The pressure was kept at 206.8 kPa. The carrier gas (argon gas) was utilized with a flow rate of 0.9 mL/min and G 2.5 volumetric gas meter (Metrex, Popayán, Cauca, Colombia) with a precision of 0.04 $m^3$/h, and a maximum working pressure of 40 kPa

was used to register the gas. For statistical analysis of hydrogen production in different fermentative condition, the *t*-test was performed using the Minitab 16 program, with 95% significant differences.

## 3. Results and Discussion

### 3.1. Hydrogen Production by Dark Fermentation

In order to verify whether the supplementation of organic wastes to coffee mucilage is feasible for hydrogen production and to determine the best operating conditions, a two level factorial design was applied to different independent factors of organic wastes ratio, chemical oxygen demand, temperature, and pH. The light independent process (anaerobic dark fermentation) principally occurs with anaerobic bacteria, which are able to grow on sources abundant in carbohydrates but not requiring light energy [9]. The Embden-Meyerhof (glycolytic pathway) is a well-known metabolic process for glucose decomposition converted into pyruvate. In this metabolism, two hydrogen atoms are released from a glucose molecule by donating electrons in the redox reaction while an oxidized nicotinamide adenine dinucleotide (NAD+) molecule is converted into a reduced form of nicotinamide adenine dinucleotide (NADH) by accepting proton from the nicotinamide ring (Equation (1)). Due to the presence of hexose sugars in coffee mucilage and carbon sources in organic wastes, these are possibly capable of converting sugars into pyruvate through anaerobic glycolysis and generating two molecules of hydrogen as by-products:

$$C_6H_{12}O_6 + 2\,NAD^+ \rightarrow 2\,CH_3COCOOH + 2\,NADH + 2\,H^+ \tag{1}$$

The two-level factorial design with IV resolution generated a total of 26 experimental runs, including two control tests. To prevent lurking variations, all designed experiments were performed in random order, and a cumulative hydrogen production was measured in each fermentation. The experiment sets, independent factors, and results of hydrogen yields are summarized in Table 1. The total cumulative hydrogen from mixed substrates varied to each different ratios; some tests promoted the formation of hydrogen within 72 h. However, less to no hydrogen yields were observed in other experiments, associated with experimental parameters of substrate ratio, Chemical Oxygen Demand (COD), temperature, and pH.

The highest hydrogen concentration of 25.9 L was achieved within 72 h at 30 °C, pH 7.0, chemical oxygen demand 60 g $O_2$/L with level 1 preparation (8 coffee mucilage: 2 organic wastes), which was equal to 1.295 L hydrogen/L substrate (Run 8 in Table 1). On the other hand, most of the tests with level 3 (2 coffee mucilage: 8 organic wastes) were not suitable for anaerobic fermentation (Runs 19–25 in Table 1), and results from 2 (5 parts coffee mucilage: 5 parts organic wastes) resulted in lower hydrogen yields (Runs 10–17 in Table 1). Even though some tests produced hydrogen in level 2 (up to 11.4 L), the fermentative digestion in the presence of 20% (*w/w*) organic wastes (level 1) was more effective compared to the other ratios. The control run without organic wastes (level 0, run 1) and the test with only organic wastes (level 4, test 26) resulted in little to no hydrogen yield, respectively. Further variance analysis of hydrogen production with the random effects model analysis of variance (ANOVA), tests showed that the high concentration of chemical oxygen demand, low temperature, and low temperature considerably influenced the final yields (Figure 1A). To obtain precise variability response with higher than 0.95 probability worth (>95% coefficient correlation, $R^2$), the weak multilateral factors were ruled out, and the accurate model was fitted with a >95% two-sided confidence interval. Individual and interaction effects of each factors were depicted in Figure 1B (Pareto Chart), presenting the similar data analysis of ANOVA tests that organic wastes, chemical oxygen demand, and temperatures were the main contributors for anaerobic hydrogen fermentation. The fitted model was generated with the major parameters in response to the reciprocal interaction of independent factors at a given condition: hydrogen production (L) = −79.70 + 0.3062 COD + 1.5725 Temperature + 14.175 pH − 0.007375 COD × Temperature + 0.02438 COD × pH − 0.2675 Temperature × pH. The optimal experimental condition for the maximal anaerobic hydrogen fermentation was

calculated and obtained though the contour plots in Figure 1C. This condition is used for anaerobic fermentative digestion for hydrogen production and is applied to fermentation with the isolated pure bacterium.

**Table 1.** A two level half-fractional factional factorial design and hydrogen production from waste substrates (coffee mucilage combined to organic wastes). Anaerobic fermentative digestion was carried out in 30 L bioreactor (20 L working volume) at provided conditions with a helical ribbon impeller mixing at 100 rpm until the hydrogen generation was completed. All runs were conducted in duplicate and provided statistical analysis of hydrogen production with 95% significant differences.

| Run | Ratio [1] | COD (g O$_2$/L) | Temperature (°C) | pH | Total Solids (g/L) | Total Volatile Solids (g/L) | Hydrogen Production (L) | Yield (L H$_2$/L Substrate) | Yield (mol H$_2$/mol Hexose) |
|---|---|---|---|---|---|---|---|---|---|
| 1 | 10:0 | 40 | 35 | 6.0 | 12.80 | 10.76 | 0 | 0 | 0 |
| 2 | 8:2 | 20 | 30 | 5.0 | 12.52 | 10.46 | 2.4 | 0.12 | 0.02 |
| 3 | 8:2 | 20 | 40 | 5.0 | 13.61 | 11.40 | 3.2 | 0.16 | 0.031 |
| 4 | 8:2 | 20 | 30 | 7.0 | 18.75 | 15.70 | 15.6 | 0.78 | 0.149 |
| 5 | 8:2 | 20 | 40 | 7.0 | 44.62 | 37.50 | 11.2 | 0.56 | 0.107 |
| 6 | 8:2 | 60 | 30 | 5.0 | 19.42 | 16.30 | 10.6 | 0.53 | 0.101 |
| 7 | 8:2 | 60 | 40 | 5.0 | 40.00 | 33.60 | 8.6 | 0.43 | 0.082 |
| 8 | 8:2 | 60 | 30 | 7.0 | 46.02 | 38.66 | 25.9 | 1.295 | 0.248 |
| 9 | 8:2 | 60 | 40 | 7.0 | 25.23 | 21.16 | 18.4 | 0.92 | 0.176 |
| 10 | 5:5 | 20 | 30 | 5.0 | 41.72 | 35.00 | 0 | 0 | 0 |
| 11 | 5:5 | 20 | 40 | 5.0 | 62.31 | 52.30 | 0 | 0 | 0 |
| 12 | 5:5 | 20 | 30 | 7.0 | 69.92 | 58.75 | 8.3 | 0.415 | 0.079 |
| 13 | 5:5 | 20 | 40 | 7.0 | 68.00 | 57.13 | 2.6 | 0.13 | 0.025 |
| 14 | 5:5 | 60 | 30 | 5.0 | 61.91 | 52.00 | 11.4 | 0.57 | 0.109 |
| 15 | 5:5 | 60 | 40 | 5.0 | 77.42 | 65.00 | 7.0 | 0.35 | 0.067 |
| 16 | 5:5 | 60 | 30 | 7.0 | 84.42 | 70.90 | 8.6 | 0.43 | 0.082 |
| 17 | 5:5 | 60 | 40 | 7.0 | 71.02 | 59.63 | 6.8 | 0.34 | 0.065 |
| 18 | 2:8 | 20 | 30 | 5.0 | 31.52 | 26.44 | 0 | 0 | 0 |
| 19 | 2:8 | 20 | 40 | 5.0 | 26.31 | 22.12 | 0 | 0 | 0 |
| 20 | 2:8 | 20 | 30 | 6.0 | 63.52 | 53.38 | 0 | 0 | 0 |
| 21 | 2:8 | 20 | 40 | 8.0 | 28.62 | 24.06 | 0 | 0 | 0 |
| 22 | 2:8 | 60 | 30 | 5.0 | 51.84 | 43.50 | 0 | 0 | 0 |
| 23 | 2:8 | 60 | 40 | 5.0 | 67.57 | 56.66 | 0 | 0 | 0 |
| 24 | 2:8 | 60 | 30 | 8.0 | 32.72 | 27.48 | 1.9 | 0.095 | 0.018 |
| 25 | 2:8 | 60 | 40 | 8.0 | 76.42 | 64.14 | 3.0 | 0.15 | 0.029 |
| 26 | 0:10 | 40 | 35 | 6.5 | 45.42 | 38.16 | 0.4 | 0.02 | 0.004 |

[1] Substrate ratio (mucilage: organic wastes).

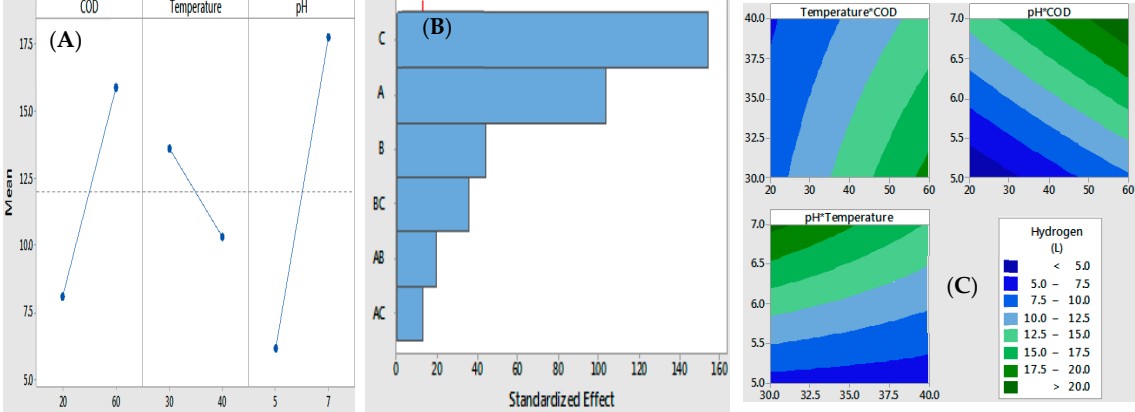

**Figure 1.** (**A**) The key effect of chemical oxygen demand, temperature, and pH on the hydrogen production from dark fermentation of complex substrates (level 1). (**B**) Individual and mutual interaction effects of chemical oxygen demand, temperature, and pH. A: chemical oxygen demand, B: temperature, C: pH. (**C**) Contour plots indicate influence of temperature and chemical oxygen demand, pH and COD, and pH and temperature, respectively. The different colors denote the different concentrations of hydrogen from dark fermentation. The dark green indicates a higher hydrogen yield while a dark blue presents a lower hydrogen yield.

### 3.2. Identification of Microorganisms

In order to identify the microorganisms which could dominantly grow and were associated with hydrogen production in complex waste substrates, a slurry sample from the best hydrogen production was spread out on a nutrient agar medium and anaerobically cultivated at room temperature. Each grown colony was suspended with a sterilized saline solution, cultivated, and biochemically reacted with provided reagents in the Vitek 2 analyzer (Biomérieux, Lyon, France). It is worthwhile to note that these isolation and identification methods would not provide all potential microorganisms in the mixed waste substrate. This work has considered isolating and identifying microorganisms that are able to primarily grow in the chosen agar medium under mesophilic and anaerobic experimental conditions (similar to bioreactors). The biochemical identification including morphology, gram staining, and species are summarized in Table 2.

**Table 2.** Isolation and biochemical analysis of microorganisms from the slurry sample from test 8.

| Code | Morphology | Color | Gram Stain | Species | Certainty (%) |
|------|-----------|-------|-----------|---------|---------------|
| B1 | Circular, entire edge, convex | Yellow | Positive | *Micrococcus luteus* | 99 |
| B2 | Irregular, irregular edge, flat | Beige | Positive | *Kocuria kristinae* | 87 |
| B3 | Circular, entire edge, flat | Beige | Positive | *Streptococcus uberis* | 87 |
| B4 | Circular, entire edge, flat | Orange | Positive | *Leuconostoc mesenteroides* ssp. *cremosis* | 94 |
| B5 | Irregular, irregular edge, convex | Beige | Positive | *Brevibacillus laterosporus* | 94 |
| B6 | Irregular, entire edge convex, | Beige | Positive | *Bacillus farraginis/smithii/fordii* | 97 |
| B7 | Circular, entire edge, convex | Beige | Positive | *Staphylococcus epudermidis* | 95 |

Biochemical identification found that seven different bacteria existed, and they were involved in the dark hydrogen fermentation under the given conditions. All identified bacteria were Gram positive, anaerobic, mesophilic, and prefer to live in soil or organic agricultural wastes such as coffee mucilage. In particular, *Streptococcu uberis* (B3) and *Leuconostoc mesenteroides* ssp. *cremosis* (B4) are capable of producing lactic acid through glycolysis (Embden-Meyerhof-Parnas pathway) using glucose. Other bacteria containing *Brevibacillus laterosporus* (B5), *Bacillus farraginis/smithii/fordii* (B6), *Micrococcus luteus* (B1), and *Kocuria kristinae* (B2) are known to metabolize complex substrates by producing hydrolytic-protease and hydrolytic-glucosidase, which allow them to contribute toward hydrogen production [40–43]. Previous works observed that they tended to undergo an acetic or butyric pathway and produce hydrogen by consuming acetic or lactic acid as substrates [44,45]. An increase of lactic and acetic acid concentration was detected in the beginning of the fermentation (up to 13.76 g/L and 5.32 g/L, respectively) while none and a lower concentration (4.13 g/L) of both acids were determined at the end of fermentation. This observation supports that these strains possibly utilize intermediate molecules (lactic acid and acetic acid) for their growth, population, and metabolism that subsequently produce hydrogen. A similar study done by Hernández et al. [34] observed that the addition of swine manure content into coffee mucilage improved the hydrogen production and methanogenic process, which is considered a major limiting factor for the hydrogen metabolic pathway. They suggested that a high carbon/nitrogen ratio (C/N) contributed toward increasing hydrogen but decreasing the methane percentage by possibly changing the metabolic pathways through dominant microorganisms and their activity [34,46,47]. They also confirmed that the C/N ratio of 53.4 had a stable hydrogen production in the repetitive batch cultivation for 140 days, which may indicate that the methanogenic pathway was inhibited during the long fermentation times by changes in metabolic routes [32,33,48]. Further 16S rDNA and sequence analysis tests confirmed that B3, B5, and B6 strains were matched (>98% similarity) with *Bacillus firmus* (KT720243.1), *Bacillus simplex* (KT922035.1), and *Frigoritolerans* (KT719834.1), respectively. For the other strains it was not possible to match their sequences with the database from the Gene bank, BLASTN (National Center for Biotechnology Information, Bethesda, MD, USA).

### 3.3. Hydrogen Production from Single Bacterium Inoculum

The effect of single bacterial fermentation on hydrogen production was studied with seven different isolated bacteria under similar experimental conditions of test 8 in Table 1. All fermentation runs were completed within 48 h, and the resulting hydrogen yields are summarized in Table 3. When isolated pure bacteria were used, the final concentration of hydrogen was in the range of 0–9.3 L, which was significantly lower than the result from the initial test without inoculum (25.9 L) (Table 3).

**Table 3.** Anaerobic dark fermentation of the complex substrate using single isolated bacteria. All fermentation was carried out with the complex substrate (8 coffee mucilage: 2 organic wastes) at 30 °C, pH 7.0, chemical oxygen demand 60 g $O_2$/L with a helical ribbon impeller mixing at 100 rpm. All tests were in duplicate and provided statistical analysis of hydrogen yields with 95% significant differences.

| Code | Species | $H_2$ Production (L) | Yield (L $H_2$/L Substrate) | Yield (mol $H_2$/mol Hexose) |
|---|---|---|---|---|
| B1 | *Micrococcus luteus* | 3.9 | 0.195 | 0.037 |
| B2 | *Kocuria kristinae* | 9.3 | 0.465 | 0.089 |
| B3 | *Streptococcus uberis* | 5.9 | 0.295 | 0.056 |
| B4 | *Leuconostoc mesenteroides* ssp. *cremosis* | 0 | 0 | 0 |
| B5 | *Brevibacillus laterosporus* | 5.6 | 0.28 | 0.054 |
| B6 | *Bacillus farraginis/smithii/fordii* | 1.8 | 0.09 | 0.017 |
| B7 | *Staphylococcus epudermidis* | 0.3 | 0.015 | 0.003 |
| B2 and B3 | *Kocuria kristinae, Streptococcus uberis* | 14.7 | 0.735 | 0.14 |

Although anaerobic fermentative digestion was able to convert carbon sources into hydrogen, we hypothesized that co-cultivation (bacterial consortium) could be responsible for enhancing hydrogen production. To prove this hypothesis, two isolated pure bacteria having the highest yield, (*Kocuria kristinae* (B2) and *Steptococcus uberis* (B3)), were inoculated for a co-cultivation fermentation. As a result, >58% higher hydrogen production ($14.7 \pm 0.8$ L, $p$-value < 0.05) was observed after 24 h, which was relatively higher than the results from single batch fermentation (Table 3). This result may suggest that these bacteria were significantly associated with hydrogen production. This study is in agreement with other earlier studies that microbial population shifts and enzymatic/metabolic shifts are critical factors for hydrogen production during dark fermentation, and these two parameters can independently or simultaneously affect the hydrogen yield [49,50]. For example, microbial sporulation, particularly in *Clostridium* sp., can be activated as a protection system when the microbial faced on un-favorite circumstance such as high temperature (>90 °C), low pH, dissolved oxygen concentration, and limitation of nutrient sources [51]. Addressing previous observations and current data, other combinations (of two or more bacteria) under different conditions could lead to better hydrogen yields via synergetic metabolisms and/or changing the pathways affecting different bacterial growth performances. The current study mainly focused on the utilization of complex substrates (coffee mucilage plus organic wastes) for hydrogen production and the determination of its optimal experimental conditions. However, the key outcome in this work is the observation that the bacterial population has a considerable effect on dark fermentation, and single batch fermentation is not suitable for efficient hydrogen production at the given conditions. This data indicates the need for further investigation with respect to the changes in bacterial growth, population, metabolic shift, microbial/product inhibition, and bioreactor classification, such as a continuously stirred tank reactor, anaerobic sequencing batch reactor, anaerobic membrane bioreactor, or immobilized bioreactor.

The current work is comparable with previous studies about the anaerobic hydrogen fermentation from other carbon sources, which include food wastes, apple, domestic wastewater, wastepaper, glycerol, and glucose; the detailed fermentation conditions and hydrogen yields are summarized in Table 4.

**Table 4.** Comparison of hydrogen yield from different carbon sources (wastes or hexose) in anaerobic batch fermentation using mixed culture or pure strain.

| Organism | Carbon Source | Reactor | Hydrogen Yield | Reference |
|---|---|---|---|---|
| Mesophilic mixed culture | Coffee mucilage + organic wastes (20%, *w*/*w*) | Batch | 0.248 mol $H_2$/mol hexose (1.295 L/L substrate) | Current work |
| Mesophilic mixed culture | Food wastes | Batch | 0.05 mol $H_2$/mol hexose | [52] |
| Mixed culture | Apple (9 g COD/L) | Batch | 0.9 L $H_2$/L substrate | [53] |
| Mixed culture | Domestic wastewater | Batch | 0.01 L $H_2$/L substrate | [53] |
| *Ruminococcus albus* | Wastepaper | Batch | 2.29 mol $H_2$/mol hexose (282.76 L/kg dry biomass) | [54] |
| *Halanaerobium saccharolyticum* | Glycerol | Batch | 0.58 mol $H_2$/mol glycerol | [55] |
| *Escherichia coli* BW25113 (engineered) | Glucose | Batch | 1.82 mol $H_2$/mol glucose | [56] |

When the carbon substrates in waste sources were fermented under mixed batch culture conditions, the result from the present study was relatively higher than those from food waste (5 times), apple (1.44 times), and the domestic wastewater (130 times). It is possible that supplementation of organic wastes is positively effective for the growth and functional activity of hydrogen-producing bacteria to enhance the yield by providing essential nutrients such as nitrogen, phosphorous, ferrous, some mineral, and metals. Previous work demonstrated that additional nitrogen from organic wastes could enhance the hydrogen yield in anaerobic digestion by increasing the C/N ratios [34]. Since nitrogen is one of the vital sources for bacterial growth, the C/N ratio affects cell growth and the metabolic pathway, which suggests a range of 6.7–47 for optimal bacterial growth [57]. Another study with phosphorous (P) demonstrated that it was an essential component for adenosinetriphosphate (ATP) formation and could develop the metabolic pathway and hydrogen production by acting in enzyme linkage for its functions [57]. On the other hand, pure microbial fermentation in the presence of waste paper or hexose obtained 2.3–9.2 times higher hydrogen yields. It is assumed that differences in the composition of substrates and microbial consortium might be highly associated with their pathway during dark fermentation. Related work of the effect of carbohydrates (mainly glucose, fructose, sucrose, and cellobiose) elucidated that the hydrogen yield decreased from 1.82 to 1.38 (mol $H_2$/mol hexose) when the chains of carbohydrates increased due to the microbial population [58]. A similar study also found that a substrate rich in carbohydrate (sucrose) was more effective in producing hydrogen when the complex substrates were used under the same experimental conditions [57,59].

It is worthwhile to highlight that the carbohydrate degradation and hydrogen production from complex substrates like coffee mucilage and organic wastes are feasible without requiring aseptic condition, nutrients, or pure cell inoculum, suggesting potential and practical application for the real industrial field. The microbial diversity and its mechanisms for engaged management, however, still remain to be discovered. The proposed model and hydrogen yield in current study would be improved with the strategies for microbial population, their gene level resources, or performing conditions (culture operation, mixing condition, reactor type, engineered strain and others).

## 4. Conclusions

Anaerobic fermentative digestion enables the hexoses in coffee mucilage combined with organic waste to be metabolized and generate hydrogen. The optimal dark fermentation conditions determined via two level factorial designs were 30 °C, pH 7.0, chemical oxygen demand 60 g $O_2$/L in the presence of 20% (*w*/*w*) organic waste, which resulted in 25.9 L hydrogen yield. Moreover, this work found that seven different bacteria were involved in the best hydrogen production test, and their individual batch fermentations under similar experimental conditions, produced lower levels of hydrogen by the end of fermentations, suggesting that further knowledge of the microbial interactions with complex

substrates and efficient anaerobic dark fermentation are required. In summary, hydrogen production from raw coffee mucilage and organic acid mixtures is feasible without any aseptic process and supplementation prior to anaerobic digestion. It can be considered a potential sources for practical hydrogen fermentation, requiring the extension of knowledge of microbial interaction to deal with complex substrates and efficient anaerobic dark fermentation.

**Author Contributions:** E.L.M.C. and A.D.Z.-Z. initiated current work at Universidad Nacional de Colombia and completed with D.K. at Hood College. As primary authors of this research, E.L.M.C. and A.D.Z.-Z. performed the designed dark fermentation tests, data collection, and literature research. D.K. assisted to review, data analysis and summary for this manuscript.

**Funding:** This research was funded by National University of Colombia, grant number 19522, 24025, and 30085.

**Acknowledgments:** The authors thank Donna Harrison and Craig Laufer at Hood College for their internal review of this work; the San Rafael farm and Central Mayorista de Antioquia for supporting coffee mucilage and organic wastes, respectively.

**Conflicts of Interest:** The authors declare no conflict of interest.

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
