# Peer review of "Hydrogen Production from Coffee Mucilage in Dark Fermentation with Organic Wastes"

_energies, doi:10.3390/en12010071_

Round 1

Reviewer 1 Report

The manuscript includes substantial work on biohydrogen production from coffee mucilage mixed with organic waste. The manuscript has been well-structured and the text is clear. However, there are a few aspects to be considered before it can be published in “Energies”.

-          Although not acute, the manuscript needs to go through language correction.

-          Lines 80-89: Please add a brief comparison between coffee mucilage and other coffee waste fractions regarding sugar and protein content. This will reinforce why using this fraction.

-          Lines 99-108: the paragraph should be modified. Words such as “feasible” entails techno-economic analysis which is absent in this study and “best” needs to be clarified that it respects to a set of parameters studied but not stated as the end-point of the research using this substrate.

-          Line 114: What do the authors mean by “not suitable for human consumption”? Wrong size/shape? Unsold? Expired?

-          C and N contents of the substrates should be made available. This will give the reader information about the C/N ratios at different conditions while mixing the substrates at different extent. This is of utmost importance considering future works in the field from my point-of-view. The authors actually mention the role of C/N ratio in Lines 278-283 while citing previous works and in Lines 338-342.

-          Section 2.2: In addition to C and N contents, the authors should provide more details about the substrates namely TS, VS and COD, and how they attained anaerobic conditions in the bioreactor.

-          Line 153: Please do not start a new sentence with a number.

-      In section 3.2. the authors should leave open the possibility that not all bacteria present in the medium have actually been able to grow in the chosen agar medium for identification.

-          Table 4: I’m assuming that when the authors state “with 95% significant differences” it corresponds to the overall group of data. How about pairwise comparisons? This can actually corroborate the choice of the bacteria for the co-culture experiments.

-          Line 310: Please add p-value.

-          In Section 3.3 the authors should make clear that other combinations (of 2 or more bacteria) could lead to better yields via synergism of different bacterial growth performances and provide the present author’s strategy as just a first/simple check-up test.

Author Response

General comments: We have addressed the specific comments on a point-by-point basis and modified the manuscript appropriately, as referred to below. We thank the reviewers for the helpful feedback and assure the reviewers that we take all manuscript revision, particularly for Energies, quite seriously. The original submission has been revised so that it concisely presents our significant research of hydrogen production from coffee mucilage in dark fermentation with organic wastes. We believe that data to support this finding are more clearly presented, thanks to careful reviews and comments of the reviewers, which we have incorporated into the revised manuscript.

Responses to Reviewers' comments:

Reviewer #1:

The manuscript includes substantial work on biohydrogen production from coffee mucilage mixed with organic waste. The manuscript has been well-structured and the text is clear. However, there are a few aspects to be considered before it can be published in “Energies”.

1)      Although not acute, the manuscript needs to go through language correction.

Reply: Thanks to the reviewer for comments. We have carefully reviewed and made corrections.

2)      Lines 80-89: Please add a brief comparison between coffee mucilage and other coffee waste fractions regarding sugar and protein content. This will reinforce why using this fraction.

Reply: Thanks for the reviewer’s constructive comments. We have added a brief composition of coffee husks, skin, pulp and mucilage and their comparisons in lines 84-89 on page 1 as follows, “Previous works identified that the coffee mucilage contained 85-91% (w/w) water, 6.2-7.4% (w/w) sugars, 4-5% (w/w) protein, and 1% (w/w) pectin substances [13,20,28,29], which were relatively higher than those obtained from other coffee by-products of husks, skin and pulp [19,30]. Furthermore, the sugars in the coffee mucilage include a high portion of the reducing sugar (63%, w/w) that facilitates the utilization of sugars to other molecules and commodities [17,31].” We believe that these explanations with corresponding references have clarified our research study in support of our experimental approach.

3)      Lines 99-108: the paragraph should be modified. Words such as “feasible” entails techno-economic analysis which is absent in this study and “best” needs to be clarified that it respects to a set of parameters studied but not stated as the end-point of the research using this substrate.

Reply: Thank you for pointing this out. We have reworded the relevant text to clarify this in lines 104-105 on page 3.

4)      Line 114: What do the authors mean by “not suitable for human consumption”? Wrong size/shape? Unsold? Expired?

Reply: The organic wastes are not suitable for human consumption because they have expired. We have added this point in lines 118-119 on page 3.

5)      C and N contents of the substrates should be made available. This will give the reader information about the C/N ratios at different conditions while mixing the substrates at different extent. This is of utmost importance considering future works in the field from my point-of-view. The authors actually mention the role of C/N ratio in Lines 278-283 while citing previous works and in Lines 338-342.

Reply: We agree with the reviewer’s comments: the C/N ratio at different conditions would provide useful information (e.g. cell growth pattern, change C/N ratio, evolution of bacteria, and further research for the pathways). Unfortunately, we do not have the C/N ratio data for each condition since we performed all of the tests (a set of 26 tests) in the single batch cultivation that was completed within 3 days. This work is quite different from the previous study in which they tested repetitive batch fermentations for 140 days (Hernández et al. 2014) to identify the changes in the C/N ratio with hydrogen and other gas production. Since our research is aimed at hydrogen production from the mixed waste substrates using two-level experimental conditions, we mainly focused on hydrogen production in different conditions (Table 1 and Figure 1), identification of potential bacteria (Table 2), and effect of each isolated bacterium under similar experimental conditions (Table 4). Following the reviewer’s suggestion, we have added some explanations accordingly in lines 283-286 on page 7. We believe these explanations have clarified our research study in support of our technical and conceptual novelty.

6)      Section 2.2: In addition to C and N contents, the authors should provide more details about the substrates namely TS, VS and COD, and how they attained anaerobic conditions in the bioreactor.

Reply: Thanks to the reviewer for the constructive comments. We have added TS, VS, and COD data in Table 1, and their analytical methods were described with corresponding references in lines 136-138 on page 3. We believe these have helped to better present the description and explanation in support of our research data more clearly.

7)      Line 153: Please do not start a new sentence with a number.

Reply: Thank you for pointing this out. We have corrected this in line 159 on page 4.

8)      In section 3.2. the authors should leave open the possibility that not all bacteria present in the medium have actually been able to grow in the chosen agar medium for identification.

Reply: We appreciate the reviewer’s thoughtful comments and suggestions. The reviewer is correct that our methods can select only mesophilic bacteria in the nutrient medium.  Following the reviewer’s suggestion, we have added these sentences in the manuscript (lines 257-260 and 283-286 on page 7). We believe that this explanation has clarified the manuscript.

9)      Table 4: I’m assuming that when the authors state “with 95% significant differences” it corresponds to the overall group of data. How about pairwise comparisons? This can actually corroborate the choice of the bacteria for the co-culture experiments.

Reply: Thanks to the reviewer for pointing this out with this impressive approach. We have compared each test with the duplicated data, and analysis of hydrogen yields were compared (lines 305-306 on page 8). Among them, B2 and B3 species, which have relatively higher hydrogen yields than others, were selected for the co-culture experiments. The reworded sentences and further comparison data are clearly presented in lines 309-310 on page 8, and Table 4 on page 9, respectively. We believe that our intention was to fully address the reviewer’s comments.

10)  Line 310: Please add p-value.

Reply: Thank you for pointing this out. We have added the p-value in line 311 on page 9.

11)  In Section 3.3 the authors should make clear that other combinations (of 2 or more bacteria) could lead to better yields via synergism of different bacterial growth performances and provide the present author’s strategy as just a first/simple check-up test.

Reply: Thank you for this feedback. We agree with the reviewer’s comments. We have added this in lines 319-322 on page 9 as follows “Addressing previous observations and current data, other combinations (of 2 or more bacteria) under different conditions in this study could lead to better hydrogen yields via synergetic metabolisms and/or changing the pathways in different bacterial growth performances.” We believe this sentence more clearly supports our experimental approach.

Reviewer 2 Report

Dear authors,

I have revised your text under journal´s rules and I am giving you my highlights and suggestions.

Line 53 - when using cursive for Latin nomenclature, it should be written according to the taxonomical standards, thus Cyanobacteria should go with capital C.

Line 64 - the URL should be written under a number, and the full text-link should be among other citations.

180/181 - I believe, that there should be 3 temperatures, the value of 250 degrees is twice, please check.

285 - There is no genus Frigotolerans. Please change it according to the genus Brevibacterium frigotolerans. (https://www.ncbi.nlm.nih.gov/Taxonomy/Browser/wwwtax.cgi?id=450367, https://www.ncbi.nlm.nih.gov/Taxonomy/Browser/wwwtax.cgi?id=450367) This corresponds to your GenBank accession number KT719834.1.

In the Table 3 and Table 4, the bacterium Leuconostoc mesenteroides spp. cremoris. Species and subspecies name should go without capital.

In the Table 5, you mention E. coli, which is GMO, please add strain.

In Conclusion, and earlier in your work, you identify the microorganisms capable of hydrogen production, however, as a result of your isolation result, I doubt that those 7 bacterial species are behind the overall best fitnesses of your anaerobic reactors. Many conditions such are isolation techniques, temperature and growth conditions can shift microbial isolates. Being in majority and easily cultivable, those microorganisms may not be the only one responsible for your results. I suggest you should rewrite your sentence in order to clarify their contribution. Also, in work as this, a whole metagenomic study should be done in order to help you with a comparison of bacterial genera with your isolates. Without that, the significance is relatively low, regarding your experiment design.

In conclusion, your work seems to be very promising, with good language standard, controls, experimental values and impact on coffee waste processing.

I hope my highlights will help you with publication processes.

Your reviewer

Author Response

General comments: We have addressed the specific comments on a point-by-point basis and modified the manuscript appropriately, as referred to below. We thank the reviewers for the helpful feedback and assure the reviewers that we take all manuscript revision, particularly for Energies, quite seriously. The original submission has been revised so that it concisely presents our significant research of hydrogen production from coffee mucilage in dark fermentation with organic wastes. We believe that data to support this finding are more clearly presented, thanks to careful reviews and comments of the reviewers, which we have incorporated into the revised manuscript.

Reviewer #2:

Dear authors, I have revised your text under journal´s rules and I am giving you my highlights and suggestions.

1)      Line 53 - when using cursive for Latin nomenclature, it should be written according to the taxonomical standards, thus Cyanobacteria should go with capital C.

Reply: Thank you for the correction; we have corrected this.

2)      Line 64 - the URL should be written under a number, and the full text-link should be among other citations.

Reply: As following the reviewer’s suggestion, we have re-written it under a number, and the full text-link (reference #12) was added to references section.

3)      180/181 - I believe, that there should be 3 temperatures, the value of 250 degrees is twice, please check.

Reply: Thanks to the reviewer for pointing this out. We have changed this in lines 186-187 on page 4.

4)      285 - There is no genus Frigotolerans. Please change it according to the genus Brevibacterium frigotolerans. (https://www.ncbi.nlm.nih.gov/Taxonomy/Browser/wwwtax.cgi?id=450367, https://www.ncbi.nlm.nih.gov/Taxonomy/Browser/wwwtax.cgi?id=450367) This corresponds to your GenBank accession number KT719834.1.

Reply: The reviewer has made an excellent observation. We have corrected this to “Brevibacterium frigotolerans” as follows the reviewer’s comment.

5)      In the Table 3 and Table 4, the bacterium Leuconostoc mesenteroides spp. cremoris. Species and subspecies name should go without capital.

Reply: Thank you for considerable comments and suggestions. We have corrected this without the capital.

6)      In the Table 5, you mention E. coli, which is GMO, please add strain.

Reply: The reviewer is right that this E. coli is GMO, which was added with the strain in Table 5.

7)      In Conclusion, and earlier in your work, you identify the microorganisms capable of hydrogen production, however, as a result of your isolation result, I doubt that those 7 bacterial species are behind the overall best fitnesses of your anaerobic reactors. Many conditions such are isolation techniques, temperature and growth conditions can shift microbial isolates. Being in majority and easily cultivable, those microorganisms may not be the only one responsible for your results. I suggest you should rewrite your sentence in order to clarify their contribution.

Reply: We agree with the reviewer’s comments: the current isolation and identification methods in this study would not provide all potential microorganisms in the mixed substrates; those mainly depend on the culture and medium conditions. However, we tried to set up the cultivation conditions as similar to the reactor conditions (anaerobic, mesophilic, and in nutrient medium) to obtain the dominant microorganisms that are able to grow and have effects on the hydrogen production under given conditions. To better understand and open the possibility for potential microorganisms, we have addressed these issues in lines 257-260 and 319-322 on pages 7 and 8, respectively.

8)      Also, in work as this, a whole metagenomic study should be done in order to help you with a comparison of bacterial genera with your isolates. Without that, the significance is relatively low, regarding your experiment design.

Reply: We appreciate the reviewer’s valuable suggestions. The reviewer is correct that further studies of a whole metagenomics work would provide useful information for isolated bacteria. Unfortunately, we do not have metagenomics data at this point since our research is aimed at hydrogen production from the mixed waste substrates using two-level experimental conditions. Based on this knowledge, we mainly focused on hydrogen production under different conditions (Table 1 and Figure 1), identification of potential bacteria (Table 2), and the effects of each isolated bacterium under similar experimental conditions (Table 4). We believe these explanations have clarified our research study in support of our technical and conceptual novelty.

9)      In conclusion, your work seems to be very promising, with good language standard, controls, experimental values and impact on coffee waste processing. I hope my highlights will help you with publication processes.

Reply: We appreciate the reviewer’s thoughtful comments and suggestions. Following the reviewer’s requests, we have addressed the comments in an exhaustive and, we believe, sound basis. We believe that our intention was to fully address all of the reviewer’s comments.

10)   Academic editor’s comment: Figure 2 is not clear. Please make it bigger and clearer.

Reply: We appreciate the reviewer’s thoughtful comments. We have made it bigger and clearer.
